# A School Mental Health Provider Like Me: Links Between Peer Racial Harassment, Depressive Symptoms, and Race-Matched School Counselors and Psychologists

**DOI:** 10.3390/ijerph22040553

**Published:** 2025-04-03

**Authors:** Sean Darling-Hammond, Cindy Le

**Affiliations:** 1Department of Community Health Sciences, University of California, Berkeley, CA 94720, USA; 2Department of Community Health Sciences, University of California, Los Angeles, CA 90095, USA; cindyle.mph@gmail.com

**Keywords:** racial harassment, depression, school psychologists, school counselors, race matching

## Abstract

Legal scholarship and caselaw suggest that exposure to peer racial harassment in school (PRHS) harms student mental health and can derail students’ academic trajectories. Legal precedents call on schools to intervene to reduce student exposure to PRHS when feasible. However, little quantitative social science has explored the impacts of PRHS, explored whether exposure to PRHS varies by racial group, or identified structural factors that may protect against PRHS. We review data from over 350,000 California 6th–12th-grade students in nearly 1000 schools and estimate that exposure to PRHS is related to a twenty-percentage-point-higher depressive symptom rate for students of all racial groups, that Black students are significantly more likely to experience PRHS, that being in a school with a race-matched school counselor or psychologist is related to lower rates of both PRHS and depressive symptoms, but that White students are more likely than students of other backgrounds to be in a school where the mental health workforce reflects their racial background. The results suggest a need to reduce exposure to PRHS, particularly for Black students, and that expanding the diversity of school mental health providers could be a pathway to protecting students against PRHS and its attendant harms.

## 1. Introduction

### 1.1. Peer Racial Harassment in Schools

According to a review of 2023 Youth Risk Behavior Survey data, one in three U.S. high school students has experienced racism in school [1]. Scholars have estimated that exposure to racism is related to poor mental health, suicide risk, and substance use for students generally and for racial minority and Black students specifically [1,2,3,4]. However, exposure to scholastic racism is uneven, and members of racial minority groups exhibit higher rates of exposure than White students [1]. Racism in school can take many forms, including school staff treating students differently due to racial bias or students engaging in harmful behavior towards peers based on race [5,6,7]. Research on scholastic racism, broadly defined, does not disentangle these unique forms of racial harm, potentially obfuscating the unique impacts of specific forms of scholastic racism and the best ways to ameliorate them [8].

Peer racial harassment in schools (PRHS) has long been the focus of legal scholarship but has only recently received attention from social scientists [9,10,11]. PRHS can be either subtle or blatant, can be verbal or physical, and may involve attacking or questioning a student’s racial identity or right to belong in a school environment [8]. It can also include racism-related physical assaults—unwanted but intentional bodily harm experienced by an individual or group of individuals because of their race [8,12]. Examples of PRHS include a student using a racist slur to refer to a fellow student, claiming a student has an inadequate or excessive connection to a racial group (e.g., “you don’t act Black enough” or “you act too Black”), or making disparaging or exclusionary comments about the racial group to which a student belongs [8].

Courts view PRHS to be sufficiently harmful to require schools to respond to instances of harassment even as doing so naturally impinges on students’ First Amendment rights [13,14]. In a precedent-setting case—Zeno v. Pine Plains Central School District—a federal trial court found, and the second circuit court of appeals upheld, that a school district owed a Black, mixed-race student 1 million U.S. dollars in money damages due to the school district’s deliberate indifference in the face of evidence that the student was experiencing peer racial harassment [15]. The student alleged that severe peer racial harassment had pushed him out of school prematurely, impacting his educational outcomes and financial opportunities. The courts agreed and, in so doing, arguably supported the notion that PRHS can cause at least 1 million U.S. dollars worth of damage to a student’s life. The case is not a singular event, and has been leveraged as precedent to support many trial court decisions to award money damages for successful claims of PRHS [13,14].

While legal scholars and institutions view PRHS as harmful, little social science research has explored the impacts, distribution, and protective factors of this form of racism. However, the scholarship on PRHS that does exist is built on the foundation established by early scholars of peer harassment in schools which found that peer harassment predicts negative educational, behavioral, and mental health outcomes [16,17,18,19]. Atop this foundation, some emerging literature explores the nature, distribution, and correlates of PRHS. Henderson et al. (2020) conducted a phenomenological study to understand—through narrative interviews—the school-based racial harassment experiences of twenty Black or African American individuals who had recently attended U.S. k-12 schools [8]. The authors identified four major emotional themes that emerged when interviewees recounted their experiences of racial harassment: anger, sadness, confusion, and distress [8]. The authors also noted that peer racial harassment led some students to feel a sense of non-belongingness. Students experiencing repeated instances of racial harassment, and riding waves of anger, sadness, and distress, could be at higher risks of developing mental health challenges such as depressive symptoms. Indeed, anger, sadness, and distress are symptoms of clinical depression [20]. Bucchianeri and colleagues (2016) evaluated data from the 2013 Minnesota Student Survey—a statewide survey of middle and high school students—and found that exposure to PRHS was more common among Black and Asian/Pacific Islander students than among White students [21]. Vance, Boyer, and Glidden (2021) found that Black and Latinx transgender youth experienced high levels of exposure to PRHS, and that, for this population, PRHS was predictive of depressive symptoms and suicidality [22]. Finally, Eisenberg and colleagues (2021) found that rates of PRHS were lower in schools that implemented school-based diversity education activities [23].

Extant research thus has provided some insight regarding the impacts, distribution, and protective factors of PRHS. However, the one study exploring the distribution of PRHS used data collected over a decade ago, before recent sociopolitical events that might have shifted the racial distribution of PRHS (such as the growth of the Black Lives Matter movement and the rise in anti-Asian hate incidents that occurred during the COVID-19 pandemic). We believe a study of the current distribution of PRHS is thus warranted. The single study exploring mental health correlates of exposure to PRHS did so for a specific population of students (Black and Latinx transgender youth). We believe there is thus value in exploring the mental health correlates of PRHS for other student populations, and for students generally. Finally, while one study explores whether diversity education might reduce rates of PRHS, studies exploring other structural interventions that might reduce PRHS could provide additional insight into how to combat this potentially pernicious phenomenon.

### 1.2. The Broad Protective Potential of School Counselors and Psychologists

The Center for Disease Control recently wrote that schools may reduce instances and impacts of scholastic racism, broadly defined, by ensuring students have access to school-based mental health providers [1]. These school staff persons are “frequently the first to see children who are sick, stressed, traumatized, act out, or hurt themselves or others” and “are trained to address students’ [mental health] needs” [24]. School counselors and psychologists—two of the most common types of school-based mental health providers—provide both direct support to students through individual meetings and general services to the school that include teaching social and emotional skills to students, providing professional development, and advising school staff about mental health.

School counselors are tasked with engaging with students and staff in ways that may have broad, school-wide effects. The American School Counselor Association writes that school counselors should “advocate for the mental health needs of all students by offering instruction that enhances awareness of mental health, appraisal and advisement addressing academic, career and social/emotional development; short-term counseling interventions; and referrals to community resources for long-term support” [25]. It further states that the instruction counselors provide should not only “proactively enhances awareness of mental health” but also “promote positive, healthy behaviors” and help “remove the stigma associated with mental health issues” [25]. In their professional development and advising roles, counselors should “educate teachers, administrators, families and community stakeholders about the mental health concerns of students, including recognition of the role environmental factors have in causing or exacerbating mental health issues, and provide resources and information” [25]. Counselors’ roles also include efforts for structural advancement, as they are to “advocate, collaborate and coordinate with school and community stakeholders to meet the needs of the whole child and to ensure students and their families have access to mental health services, recognize and address barriers to accessing mental health services and the associated stigma, including cultural beliefs and linguistic impediments,” and “advocate for ethical use of valid and reliable universal screening instruments with concerns for cultural sensitivity and bias” [25].

The activities of school psychologists may also have school-wide impacts. The National Association of School Psychologists describes school psychologists as “uniquely qualified members of school teams that support students’ ability to learn and teachers’ ability to teach” [26]. They state that school psychologists should “partner with families, teachers, school administrators, and other professionals to create safe, healthy, and supportive learning environments that strengthen connections between home, school, and the community” [26]. They write that school psychologists should “provide direct support and interventions to students, consult with teachers, families, and other school-employed mental health professionals (i.e., school counselors, school social workers) to improve support strategies, work with school administrators to improve school-wide practices and policies, and collaborate with community providers to coordinate needed services” [26].

That school-based mental health providers take such a broad approach to enhancing mental health suggests that the presence of school-based mental health providers may confer broad, school-wide mental health benefits to members of school communities. For example, by helping students process traumatic events, they can help students avoid maladaptive behaviors, reducing the likelihood that these students engage in peer-to-peer bullying and reducing school-wide victimization. And by providing school-wide social and emotional instruction, mental health providers may enhance students’ social and emotional skills, potentially shifting students’ behaviors in ways that augment students’ abilities to foster positive relationships and improving school climate for all. Similarly, by helping teachers adopt beliefs and master practices that improve behavior management and reduce inequities in how teachers treat students of varied backgrounds, mental health providers may reduce student exposure to exclusionary discipline, and may also help reduce racial disparities in exclusionary discipline. This, too, may confer school-wide benefits.

Research has indeed found the presence of school counselors and psychologists is related to broad school-wide outcomes, including better school climates and lower expulsion rates [27,28,29,30]. In one study, authors found that schools with responsive counseling services had lower suspension rates, schools with higher school-counselor-to-student ratios had lower suspension rates and saw fewer disciplinary incidents, and that schools where principals rated the mental health provider teams as being of higher quality saw higher graduation rates and lower expulsion rates [31]. Given documented links between positive school climates and depression, and between exposure to exclusionary discipline and depression, it is conceivable that school-based health providers may reduce overall depression rates within schools by enhancing exposure to positive climates and reducing exposure to exclusionary practices [32,33]. Research has also found that school psychologists can implement universal prevention programs that reduce depression rates [34].

Prior research has thus demonstrated that school-based mental health providers can confer school-wide benefits. Thus, in this research, we conceptualize school counselors and psychologists as having broad potential impacts on school-wide behavioral and mental health outcomes. Here, we therefore explore how school counselors and psychologists might exert a broad, school-wide correlational influence on peer racial harassment and depressive symptom rates.

### 1.3. The Protective Potential of a School Mental Health Provider “Like Me”

Research has documented that exposure to school-based mental health providers is related to a variety of scholastic benefits. However, school-based mental health providers vary, and so too may their impact. A growing body of work in the race-matching tradition has demonstrated that students of color often benefit from being in schools with school staff who match their racial background, and that they benefit from being in schools characterized by more staff diversity [35,36,37,38]. Recent work has found that Black students who are demographically matched with their teachers exhibit warmer student–teacher relationships, experience fewer absences and suspensions, and perform better academically [35,36,37]. Research on race matching has expanded beyond the domain of the student–teacher relationship, and extant research suggests that having a therapist who is race-matched may lead to more productive therapeutic experiences [39]. Yet, no research explores whether students in schools with race-matched school mental health providers have better mental health outcomes.

There are many routes by which being in a school that employs at least one race-matched school-based mental health provider might exert a more positive protective influence against depression, or against peer racial harassment. Some routes flow through direct interactions between mental health providers and race-matched students. However, others flow through the school-wide impacts of school mental health providers.

Regarding direct routes, as noted above, the presence of a race-matched school psychologist could lead to a more productive therapeutic relationship in direct service contexts, protecting against or facilitating treatment of depressive symptoms [39]. In addition, the presence of a race-matched school-based mental health provider might lead to the presence of a trusted adult at school. Research has found that race-matching often leads to warmer and richer student–staff relationships and that having a trusted adult at school is a prominent protective factor against depression [35,36,40]. While these direct routes are certainly feasible, they are not the only pathways by which the presence of at least one race-match mental health provider might benefit students. Instead, race-matched school mental health providers could improve racially minoritized students’ outcomes through “school-wide” impacts.

As noted above, we conceptualize school mental health providers as having broader roles within schools that allow them to shift teacher and student behaviors in ways that may benefit students from racially minoritized backgrounds. Rooted in theoretical models related to mental health promotion in schools, enhancing school climate, harnessing the power of intergroup contact, and the ways school professionals can operate as structural determinants of student health, we further posit that school-wide routes might exert broader, school-wide impacts that benefit students from racially marginalized backgrounds [41,42,43,44].

For example, race-matched school-based mental health providers may be more culturally attuned to the unique mental health landscapes and experiences of students of their racial backgrounds. They therefore may be better equipped to provide professional development to teachers that allows these teachers to embrace culturally competent approaches that benefit students who share racial identities with these mental health providers. This could take the form of empowering teachers to identify and intervene in situations where students are being racially harassed, or it could involve helping teachers learn ways to engage with students in culturally competent ways that allow for stronger relationships. Relatedly, in Delphi studies, school psychologists have expressed that cultural competence is a critical driver of the effectiveness of their work [45,46,47,48]. School psychologists and scholars have opined that improving school psychologists’ cultural competence is a critical policy priority [49,50].

In addition to routes that flow through teacher behavior, school mental health providers of a given race may also be better able to help students avoid problematic behaviors that negatively impact members of that race, such as racial harassment. Imagine, for example, a White student who has used a racist slur against an Asian American student, and who then is required to have a conversation with their school psychologist who is Asian American. The school psychologist may be better equipped to share their lived experience as an Asian American person in ways that help that White student understand the harmfulness of the racial slur they used, empathize, and avoid engaging in racial harassment in the future. This may all redound to a protective benefit for Asian American students in that school. Thus, mental health providers may help protect students who share their racial backgrounds by expanding student empathy for members of that racial group.

### 1.4. The Present Research

The present research aims to close research gaps related to our understanding of the distribution and correlates of peer racial harassment in schools (PHRS), and the potential protective benefits of race-matched school-based mental health providers, by answering the following questions:How do rates of PRHS vary depending on students’ racial backgrounds?Is PRHS related to depressive symptoms for students of various racial backgrounds?Do students of certain backgrounds experience higher levels of exposure to race-matched school counselors and psychologists?Is the presence of a race-matched school counselor or psychologist related to lower rates of PRHS or depressive symptoms?

## 2. Materials and Methods

We leveraged data from the California Health Kids Survey (CHKS) and California Survey of School Staff (CSSS) from the 2022–2023 school year. The analyses reviewed data from 355,490 sixth–twelfth-grade students in California middle and high schools. All analyses were conducted in STATA version 18, and all visualizations were produced using Microsoft Excel.

To explore our first research question, we reviewed the rates of exposure to each of four categorical levels of racial harassment in the past twelve months (I have been harassed by a student due to my race or ethnicity: 1 = zero times, 2 = one time, 3 = two to three times, 4 = four or more times) disaggregated by students’ self-described racial backgrounds as indicated in a “select all that apply” question format (American Indian/Alaskan Native, Asian, Black or African American, Hispanic or Latinx, Native Hawaiian/Pacific Islander, or White). We visualized students’ levels of exposure to racial harassment, disaggregated by racial group, using stacked bar charts. We next dichotomized the racial harassment factor variable (such that 0 = no racial harassment, 1 = some racial harassment) and conducted an analysis of variance (ANOVA) to explore if mean levels of exposure to racial harassment differ depending on students’ self-described racial backgrounds. We conducted a hypothesis test against the null hypothesis that the ANOVA F-statistic is equal to 0, and rejected the null only if the *p*-value associated with the observed F-statistic was less than 0.05.

For our second question, we conducted sub-analyses for each racial group in which we regressed a dichotomous indicator of self-reported depressive symptoms against the dichotomized measure of racial harassment, controlling for parental education, gender, transgender status, sexual orientation, student grade level (e.g., “sixth grade”), and school of attendance. Formally, our model isDEPRESSION = α + β_1_ RACIAL HARASSMENT + β_i_X_i_(1)

Here, DEPRESSION is our dichotomized indicator of self-reported depressive symptoms, RACIAL HARASSMENT is our dichotomized indicator of self-reported exposure to peer racial harassment in school, and X_i_ is a vector of covariates. For each racially disaggregated regression analysis, we conducted a hypothesis test against the null hypothesis that β_1_ = 0, and evaluated statistical significance using the standard benchmark that the *p*-value associated with β_1_ must be less than 0.05 to reject the null.

To evaluate our third question, we first leveraged CSSS data to create a school-level dataset indicating whether, in the 2022–2023 school year, each school employed at least one school staff person who described their role at the school as being a “counselor, psychologist”, and a series of indicators regarding whether they employed a counselor or psychologist who was American Indian/Alaska Native, Asian, Black or African American, Hispanic or Latinx, Native Hawaiian/Pacific Islander, or White. Because schools were coded based on whether they employed a school mental health provider at all, or of a given race, our variables do not account for the number of mental health providers in a school. A school with four mental health providers (three White, one Black) would thus be coded as having a mental health provider (1), not having an American Indian/Alaska Native mental health provider (0), not having an Asian mental health provider (0), having a Black mental health provider (1), not having a Latinx mental health provider (0), not having a Native Hawaiian/Pacific Islander health provider (0), and having a White mental health provider (1). Once we had created these school-level indicators of whether a school employed a mental health provider, and whether the school employed mental health providers of various races, we next linked our student-level data from CHKS to these school-level data from CSSS to identify whether each student was in a school with at least one school mental health provider, and whether they were in a school with at least one race-matched mental health provider. We then measured the proportion of American Indian/Alaska Native, Asian, Black or African American, Hispanic or Latinx, Native Hawaiian/Pacific Islander, or White students who are (1) in a school with a school counselor or psychologist and (2) in a school with a race-matched school counselor or psychologist and presented results visually in a simple bar chart.

Importantly, when we say that a student has a “race-matched” school counselor or psychologist, we simply mean that the student attends a school with a school counselor or psychologist who shares their racial identity. We do not imply that the student has a direct therapeutic relationship with that school counselor or psychologist. As previously discussed (see the Introduction), these methods reflect our conceptualization of school counselors and psychologists as school actors with broad potential influence on the culture, behaviors, beliefs, norms, and outcomes of the entire school in which they work.

To evaluate our fourth research question, for students of each racial group, we calculated the correlation between racial harassment and having a school counselor or psychologist of each racial background. Thus, for example, among Asian students, we calculated the correlation between racial harassment and having an Asian school mental health provider, then calculated the correlation between racial harassment and having a Black school mental health provider, and so on; and then we repeated this sequence for Black students, Hispanic students, and so on. We used bar charts to visualize these correlations. To empirically ascertain if school mental health providers of a given race might confer unique protective benefits for students who share their racial identities, we calculated Jenrich χ^2^ values and corresponding *p*-values for the null hypothesis that race-matched and non-race-matched correlations are equivalent using STATA’s mvtest command. This command line, for example, quietly calculates the correlation between having a Black mental health provider and racial harassment *among Black students*, quietly calculates the correlation between having a Black mental health provider and racial harassment *among non-Black students*, and compares the two to see if these correlations are statistically significantly distinct from one another. The resulting comparison indicates whether the protective potential of exposure to a Black mental health provider is stronger for Black students than for non-Black students and can be considered a form of moderation analysis.

We repeated this analysis for students of other racial identities. In all cases, we rejected the null hypothesis if the *p*-value associated with a given Jenrich χ^2^ value was less than 0.05. Having completed analyses related to racial harassment, we then calculated Jenrich χ^2^ values, and corresponding *p*-values, to compare the correlation between having a school counselor or psychologist of a given racial group and *depressive symptoms* for students who do, and do not, share the racial identity of their school mental health provider. We again rejected the null hypothesis if the *p*-value associated with a given Jenrich χ^2^ score was less than 0.05.

Across tests, we did not make a correction for multiple tests because (as seen below) our *p*-values are universally too low for such a correction to augment our findings. We did not control for the number of mental health providers in a school due to the natural collinearity between the number of mental health providers in a school and whether that school employs a mental health provider of a specific racial background.

## 3. Results

Below, we summarize the demographic characteristics of our sample. Thereafter, we discuss the findings from our analyses.

### 3.1. Sample Characteristics

The demographics of our sample of 355,519 California middle and high school students largely mirror those of the state of California. A plurality of the students in our sample identified as Hispanic/Latinx (50.4%), and sizeable percentages identified as White (20.3%) or Asian (17.4%). Black (8.5%), American Indian/Alaska Native (4.2%), and Native Hawaiian/Pacific Islander (2.8%) students comprised smaller shares of the sample. Approximately one in seven students (15%) indicated having been racially harassed, and approximately one in three (33.2%) indicated experiencing depressive symptoms. Most students (84.8%) attended a school with at least one school counselor or psychologist (and while we do not calculate or report it here, many attended a school with more than one). Of those who attended a school with a counselor or psychologist, a majority attended a school with at least one White (56.8%) or Hispanic/Latinx (51.2%) school counselor or psychologist. Smaller percentages attended a school with at least one Asian (20.4%), Black (13.9%), American Indian/Alaska Native (3.3%), or Native Hawaiian/Pacific Islander (2.3%) school counselor or psychologist. See Table 1 for additional sample demographics.

### 3.2. Rates of Racial Harassment Differ by Race

As depicted in Figure 1, Black students are markedly more likely than students of any other racial background to have experienced racial harassment and are between 1.5 and 4 times more likely than students of other races to have experienced four or more instances of racial harassment. Overall, 15% of students experienced PRHS. Levels of exposure varied depending on whether students were Black (28%), Native Hawaiian/Pacific Islander (21%), Asian (20%), American Indian/Alaska Native (19%), Hispanic (11%), or White (9%). Our ANOVA empirically echoed this point (F = 1510.14, df = 5, *p* < 0.001).

### 3.3. Racial Harassment Is Related to Depressive Symptoms for Students of All Backgrounds

In each of our racially disaggregated, confounder-adjusted regression models, we found that exposure to racial harassment significantly (*p* < 0.001) predicted higher likelihoods of exhibiting depressive symptoms (Table 2). For example, after controlling for student, parent, and school characteristics (including student gender, student sexual orientation, student grade level, parental education, and school of attendance), Black students who indicated being racially harassed had a 0.23 proportion (or 23 percentage point) higher depressive symptom rate than Black students who indicated they had not been racially harassed in the last year. Certain demographic characteristics also predicted higher depressive symptom rates, including being female (versus male), transgender, gay, or bisexual (versus straight), or having parents who did not graduate from high school (relative to having parents who graduated from college).

### 3.4. White Students Are More Likely to Be in a School That Employs at Least One Race-Matched School Counselor or Psychologist

As depicted in Figure 2, generally, between 80 and 90% of students of any given racial background have at least one counselor or psychologist in their school. However, the proportion who have at least one *race-matched* school counselor or psychologist varied by student race. While 72% of White students have at least one race-matched school counselor or psychologist, percentages are far smaller for Native Hawaiian/Pacific Islander (3%), American Indian/Alaska Native (3%), Black (20%), and Asian (28%) students. Notably, more than half of Hispanic students (57%) have at least one race-matched school counselor or psychologist.

### 3.5. For Asian, Black, Hispanic, and White Students, Access to a Race-Matched School Counselor or Psychologist Is Negatively Correlated with Both Racial Harassment and Depressive Symptoms

Exposure to a school counselor or psychologist was more negatively related to racial harassment than exposure to a non-race-matched school counselor or psychologist (Figure 3). For Asian students, having an Asian school counselor or psychologist was more strongly correlated with lower rates of racial harassment than having a school counselor or psychologist of other backgrounds (Jenrich χ^2^ = 94.86, *p* < 0.001). The same was true for Black (Jenrich χ^2^ = 53.83, *p* < 0.001), Hispanic (Jenrich χ^2^ = 78.54, *p* < 0.001), and White (Jenrich χ^2^ = 130.70, *p* < 0.001) students. It was not true, however, for American Indian/Alaska Native or Native Hawaiian/Pacific Islander students, perhaps owing to the smaller proportion of American Indian/Alaska Native or Native Hawaiian/Pacific Islander students (3%) who had a race-matched school counselor or psychologist and perhaps reflecting inadequate statistical power to accurately detect the relationship in these populations. A similar, albeit tempered, trend appears in analyses regarding depressive symptoms, where the correlation between having a school counselor or psychologist of a given racial background and depressive symptoms was a function of the demographics of students such that race-matched correlations were generally stronger than non-race-matched correlations for Asian (Jenrich χ^2^ =14.30, *p* < 0.001), Black (Jenrich χ^2^ = 7.91, *p* = 0.005), Hispanic (Jenrich χ^2^ = 7.38, *p* = 0.007), and White students (Jenrich χ^2^ = 10.97, *p* < 0.001).

## 4. Discussion

This research echoes prior research suggesting that exposure to racism in school is harmful [1,2,3,4]. However, unlike much prior school-based research, this research specifically investigates peer racial harassment in schools—a form of racial harassment that has long been viewed by legal institutions and legal scholars as being uniquely harmful and has only recently received attention from social scientists [8,9,10,11,13,14,15].

This research adds to prior research by demonstrating remarkable consistency in the quantitative relationship between exposure to peer racial harassment in schools and depressive symptoms across six racial groups, documenting that across racial groups, students who have experienced racial harassment in school appear to be at least twenty percentage points more likely to exhibit depressive symptoms. It also provides evidence that the distribution of exposure to peer racial harassment in schools is not uniform, and that members of certain racial groups are more likely to be racially harassed by peers. Specifically, more than one in four Black students indicated that they had experienced peer racial harassment in school, and Black students were more than two times more likely than Hispanic students, and more than three times more likely than White students, to indicate being racially harassed. While Black students were significantly more likely to experience peer racial harassment in school than members of any other racial group, approximately one in five Native Hawaiian/Pacific Islander and Asian students also indicated being racially harassed, which is notably higher than rates for White and Hispanic students. These rates may suggest that recently documented increases in anti-Asian American Pacific Islander (AAPI) hate incidents are exerting ripple effects in schools [51].

This research also echoes prior research that has suggested that exposure to race-matched school staff and race-matched mental health professionals may provide unique benefits [35,36,37,38,39]. Specifically, we find that—for Asian, Black, Hispanic, and White students—being in a school that employs at least one race-matched school mental health provider is uniquely protective against racial harassment and depressive symptoms. We also find, however, that the distribution of exposure to race-matched school-based mental health providers is not uniform; whereas 72% of White students and 57% of Hispanic students are in schools where they enjoy a race-matched school counselor or psychologist, much smaller shares of Asian (28%), Black (20%), American Indian/Alaska Native (3%), and Native Hawaiian/Pacific Islander (3%) students attend schools with a race-matched school mental health provider.

Critically, this research is not causal in nature, and we do not claim here that exposure to peer racial harassment causes declines in mental health, nor that exposure to race-matched school-based mental health providers will necessarily reduce exposure to racial harassment or rates of depression. However, we believe that our confounder-adjusted models provide persuasive evidence that such causal effects might exist. Additional research leveraging causal identification strategies (such as randomized controlled trials or instrumental variable approaches) is needed. Another important limitation of this research is that it leverages exposure data at the school level. These data are attuned to exploring how the school-wide ripple effects created by the actions of mental health providers impact students of varied backgrounds. However, these data are not designed to answer questions about the impacts of direct student–provider experiences. Future research leveraging student-level exposure data could explore whether race-matched therapeutic experiences in schools are related to lower depressive symptom or racial harassment rates. This research could also explore whether students who interact with race-matched mental health providers exhibit a smaller relationship between exposure to PRHS and mental health outcomes.

This research suggests that race-matched school mental health providers may protect students from PRHS and depressive symptoms. Assuming this is so, future research could explore *why* this race-matching confers unique benefits and could investigate whether professional development designed to help school mental health providers develop rich relationships with, and expand empathy towards, students of all backgrounds could enhance the abilities of providers of all backgrounds to protect students of all backgrounds from PRHS and depression.

Our results suggest that school mental health providers may exert school-wide impacts that improve conditions for students who share their racial identities. Particularly given the dearth of empirical work regarding structural interventions to reduce student exposure to peer racial harassment, this research suggests that schools hoping to reduce harassment may benefit from exploring opportunities to expand the diversity of their school mental health workforce. These schools may also benefit from ensuring diverse school mental health providers are empowered to exert school-wide impacts, for example by providing professional development that can expand empathy for diverse students, and by providing social and emotional instruction to students that may make racial harassment less common.

While we find that students experience unique benefits from being in a school with a race-matched mental health provider, it is important to note that our results do not indicate that students *only* benefit from having a race-matched school mental health provider. Indeed, initial exploratory models echoed that students in schools with a mental health provider generally had lower depressive symptom and racial harassment rates, suggesting that the presence of a mental health provider, of any background, is better than the absence of one. In addition, we see that for Black students, while having a Black mental health provider was uniquely protective, having a Hispanic or Asian school mental health provider was also protective. This may suggest that diversifying the school mental health provider workforce could improve conditions for students who belong to various racially minoritized groups. It may also suggest that, regardless of the demographic composition of a school’s mental-health-providing workforce, school leaders might enhance the protective impact of mental health providers by ensuring these providers both have a high degree of cultural competence and have the skills and training needed to expand cultural competence throughout the school and ensure students and staff experience empathy for, and form strong connections with, students of all backgrounds [45,46,47,48,49,50].

Strategies related to school mental health professionals may prove impactful, but we do not suggest that they are the only strategies for combatting peer racial harassment in schools. The U.S. Department of Justice Civil Rights Division and the U.S. Department of Education Office for Civil Rights recently outlined practical strategies for reducing COVID-19-related harassment towards Asian American and Pacific Islander students. These strategies include immediately notifying a school leader when a suspected discriminatory incident occurs (ostensibly to empower timely corrective measures), writing down the details of incident (ostensibly to empower future legal action or community advocacy), having the school translate the details of the incident into other appropriate languages (e.g., the language of victim or their families, the language of the reporter, ostensibly to broaden the set of adults who can intervene, advocate, and support), and filing a formal complaint with the Civil Rights Division of the U.S. Department of Justice (civilrights.justice.gov, accessed 29 March 2024) or the Office for Civil Rights at the U.S. Department of Education (https://www.ed.gov/laws-and-policy/civil-rights-laws/file-complaint, accessed on 29 March 2024) [52]. While it is unclear whether any federal units will continue to receive and respond to claims of peer racial harassment in schools, current law provides a private right of action allowing students who have suffered from peer racial harassment in schools to bring claims of deliberate indifference against their school districts. And the action or inaction of any governmental entity does not foreclose opportunities for community advocacy. The acts of communicating proactively with educational leaders, carefully documenting evidence, and translating details into appropriate languages may buttress private legal claims and community advocacy efforts.

The Center for Disease Control has also provided guidance on reducing scholastic racism, noting that scholastic racism may in some instances be driven by scholastic cultures that punish students inequitably and suggesting that one approach may be to create disciplinary paradigms that demonstrate that all students, regardless of their social group, are valued members of school communities [1]. It writes that this may be achieved by providing professional development to staff throughout the school to increase their awareness of racism and bias and building their skillsets to intervene when scholastic racism occurs [1]. Finally, it recommends implementing strengths-based programs to help students nurture resilience to weather the racism that at least some will inevitably experience and supporting the creation of student affinity groups that can create social networks of support to help students cope with challenging instances of scholastic racism [1].

Placing this guidance in conversation with the research we present here, we argue that efforts to combat peer racial harassment may be most effective when they focus on ensuring schools have the personnel, structures, and systems needed to ensure that students and staff can treat students of all backgrounds as valued members of the school community, that staff can treat students of all backgrounds with equity and dignity, that staff can identify peer racial harassment when it occurs, and that students and staff can respond to and grow from incidents of racial harassment in ways that both repair harm and avoid repeating harmful mistakes. As the federal government restructures the Office of Civil Rights and abandons its long-standing focus on reducing racial inequity in schools, it will be incumbent on scholars and advocates to identify, elevate, and demand the implementation of promising approaches to protecting students from racial harassment, and it will be up to state education agencies, school districts, and schools to implement, refine, and scale these practices.

## 5. Conclusions

We find evidence that peer racial harassment is impactful, widespread, and unequally distributed, more often impacting Black students. Moreover, we find that while race-matched school mental health providers can protect against racial harassment and depressive symptoms, Black, Asian, and Native-Hawaiian/Pacific Islander students are less likely to enjoy access to a race-matched school mental health provider. Increasing the diversity and cultural competence of our school mental health providers, and augmenting their opportunities to provide professional development and social and emotional lessons that expand empathy and belonging, may thus prove effective strategies for reducing peer racial harassment for those most vulnerable. While we hope this research will encourage schools to explore diversifying their school mental health provider workforce, we also hope it will encourage critical thought about how best to empower school mental health providers and other school staff to protect the mental health of marginalized students. And we hope this research will encourage schools to employ, and will encourage scholars to evaluate, a variety of approaches to reducing peer racial harassment and protecting and enhancing the mental health of *all* of our nation’s youth.

## Figures and Tables

**Figure 1 ijerph-22-00553-f001:**
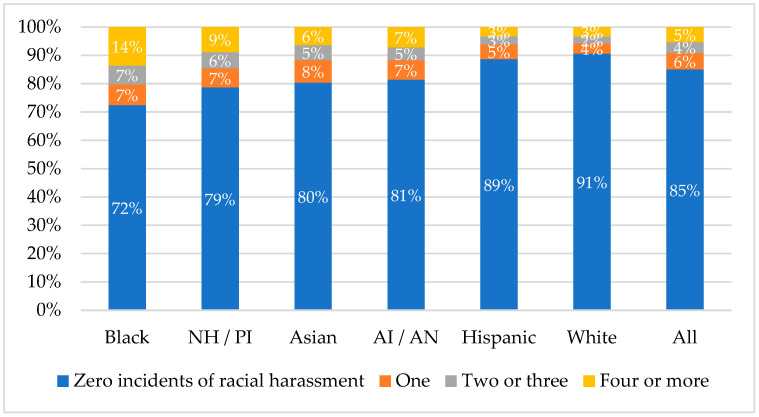
Rate of exposure to varying levels of racial harassment, disaggregated by student race. Note. NH/PI = Native Hawaiian/Pacific Islander, and AI/AN = American Indian/Alaska Native.

**Figure 2 ijerph-22-00553-f002:**
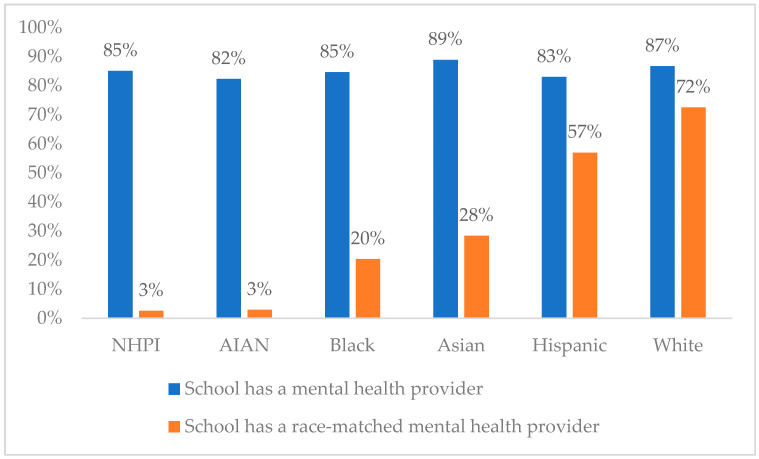
Percentage of students who have access to a school counselor or psychologist and a race-matched school counselor or psychologist, by student race. Note. NHPI = Native Hawaiian/Pacific Islander. AIAN = American Indian/Alaska Native.

**Figure 3 ijerph-22-00553-f003:**
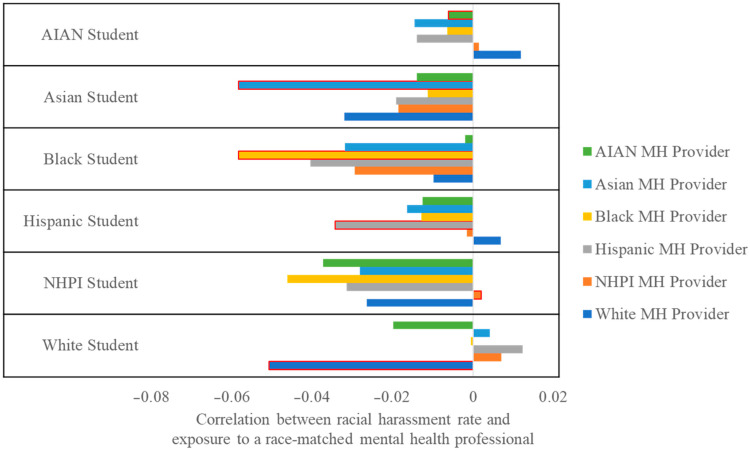
Correlation between experiencing racial harassment and having at least one school counselor or psychologist of a certain racial background, by student race. Note. Bars depict Pearson’s correlations between experiencing racial harassment and having a school counselor or psychologist of a given background, for students of specific racial backgrounds. Bars with red borders represent the relationship when the student attends a school with at least one race-matched mental health (“MH”) provider. For example, among Black students, the correlation between having a Black school counselor or psychologist and racial harassment (the yellow bar with the red border) was approximately −0.06. Meanwhile, among Black students, the correlation between having a White school counselor or psychologist and racial harassment (the blue bar with no border) was approximately −0.01—a substantially smaller absolute correlation. AIAN = American Indian/Alaska Native. NHPI = Native Hawaiian/Pacific Islander. MH = Mental Health.

**Table 1 ijerph-22-00553-t001:** Mean characteristics for a sample of 355,519 California secondary school students.

Racially harassed	15.0%
Experienced depressive symptoms	33.2%
Student race	
American Indian/Alaska Native	4.2%
Asian	17.4%
Black	8.5%
Hispanic	50.4%
Native Hawaiian/Pacific Islander	2.8%
White	20.3%
Grade	
6th grade	3.7%
7th grade	28.2%
8th grade	5.1%
9th grade	27.5%
10th grade	6.5%
11th grade	24.4%
12th grade	4.4%
Female	46.8%
Transgender	2.2%
Sexual orientation	
Straight	76.8%
Gay	2.6%
Bisexual	8.4%
Parental education	
Did not finish HS	12.3%
Graduated HS	16.5%
Attended college	10.4%
Graduated college	41.3%
Has a school counselor or psychologist	84.8%
Demographics of students’ school counselor or psychologist (among students with a counselor or psychologist)	
American Indian/Alaska Native	3.3%
Asian	20.4%
Black	13.9%
Hispanic	51.2%
Native Hawaiian/Pacific Islander	2.3%
White	56.8%

**Table 2 ijerph-22-00553-t002:** Likelihood of indicating having depressive symptoms as a function of racial harassment and student-, parent-, and school-level controls, by six racial subgroups.

	American Indian/Alaska Native	Asian	Black	Hispanic	Native Hawaiian/Pacific Islander	White
Racial harassment	0.22 *** (0.01)	0.22 *** (0.00)	0.23 *** (0.01)	0.24 *** (0.00)	0.23 *** (0.01)	0.23 *** (0.00)
Female ^#^	0.17 *** (0.01)	0.12 *** (0.00)	0.14 *** (0.01)	0.16 *** (0.00)	0.15 *** (0.01)	0.13 *** (0.00)
Transgender ^^^	0.12 *** (0.03)	0.16 *** (0.01)	0.08 *** (0.02)	0.015 *** (0.01)	0.12 *** (0.03)	0.19 *** (0.01)
Gay ^^^	0.19 *** (0.02)	0.19 *** (0.01)	0.18 *** (0.01)	0.20 *** (0.01)	0.16 *** (0.02)	0.24 *** (0.01)
Bisexual ^^^	0.25 *** (0.01)	0.23 *** (0.01)	0.22 *** (0.01)	0.25 *** (0.00)	0.24 *** (0.02)	0.26 *** (0.00)
Parents graduated HS ^+^	−0.01 (0.01)	−0.02 ** (0.01)	−0.01 (0.01)	−0.02 *** (0.00)	−0.02 (0.02)	−0.01 (0.01)
Parents attended some college ^+^	0.00 (0.01)	−0.00 (0.01)	−0.01 (0.01)	−0.01 * (0.00)	−0.01 (0.02)	−0.01 (0.01)
Parents graduated from college ^+^	−0.05 *** (0.01)	−0.06 *** (0.01)	−0.04 *** (0.01)	−0.04 *** (0.00)	−0.05 ** (0.02)	−0.07 *** (0.01)
Grade	--	--	--	--	--	--
School	--	--	--	--	--	--
N	17,444	85,001	34,592	214,393	12,099	148,694

^#^ Reference category is “male”. ^^^ Reference category is “straight”. ^+^ Reference category is “did not graduate high school”. * *p* < 0.05, ** *p* < 0.01, *** *p* < 0.001.

## Data Availability

All data used in this research, including survey results from the California Healthy Kids Survey (CHKS) and California Survey of School Staff (CSSS), can be accessed via application to, and subsequent purchase from, WestEd, Inc.

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
