# Peer review of "A School Mental Health Provider Like Me: Links Between Peer Racial Harassment, Depressive Symptoms, and Race-Matched School Counselors and Psychologists"

_ijerph, 2025, doi:10.3390/ijerph22040553_

Round 1

Reviewer 1 Report

Comments and Suggestions for Authors

Thank you for the opportunity to review the manuscript titled “A School Mental Health Provider Like Me: Links Between Peer Racial Harassment, Depressive Symptoms, and Race-Matched School Counselors and Psychologists.” Unfortunately, I could not access the supplementary materials because the link in the manuscript does not work.

This paper presents an original cross-sectional study investigating the distribution and correlates of peer racial harassment, along with the potential protective benefits of race-matched school-based counselors and psychologists. The researchers utilized data from the California Healthy Kids Survey and California Survey of School Staff from the 2022-23 school year, analyzing data from 355,490 sixth to twelfth-grade students across California middle and high schools.

The results highlight that rates of racial harassment vary by race; Black students are significantly more likely than students of other racial backgrounds to have experienced racial harassment. Furthermore, racial harassment is associated with depressive symptoms in students of all backgrounds; White students are more likely to encounter a race-matched school counselor or psychologist; for Asian, Black, Hispanic, and White students, access to a race-matched school counselor or psychologist is inversely correlated with both racial harassment and depressive symptoms.

The title accurately reflects the main topic of the paper. While the keywords are suitable, the abstract could be improved (see my comments below).

The introduction is well-structured and well-written, providing a comprehensive summary of the relevant literature and offering sufficient background for non-specialist readers to understand the study’s purpose. However, it could be enhanced (see my comments below). The rationale for the study is adequately explained, and the objectives are clearly outlined in the main text through four research questions.

The methods employed are rigorous, well-explained, and appropriate for the study’s aims. Sufficient information is provided for a competent researcher to replicate the survey and statistical analyses, as the methods and instruments are presented clearly. The results are presented clearly in an appropriate format, with tables and figures effectively summarizing essential data that could not be easily included in the text, making them easy to interpret. Appropriate statistical methods were used to address the study’s aims. All possible interpretations of the data were considered, and no alternative hypotheses consistent with the available data were ignored.

The discussion is well-written and easy to follow, though it could be improved (see below). The number and scientific quality of the references cited are adequate and up-to-date; however, they could be enhanced by addressing the comments I made regarding the introduction.

In conclusion, the study is timely, and interesting, and the manuscript is well-written. The research methodology is rigorous, and the findings are intriguing and significant. While the article should be considered for publication in the International Journal of Environmental Research and Public Health, I have some recommendations for improving the manuscript. Below, I provide comments on the manuscript page by page, rather than ranking them by importance.

To improve the coherence and accuracy of the manuscript, here are the suggested revisions for each section:

Abstract: In the abstract, it is stated that Black, Asian, and Native Hawaiian/Pacific Islander students are significantly more likely to experience peer racial harassment. This is not consistent with what is said in the main text, i.e., that rates of racial harassment differ by race and that Black students are markedly more likely than students of any other racial background to have experienced racial harassment. Please revise the abstract accordingly.

Introduction: While it is true that racial harassment in school has been the focus of legal scholarship, over the last twenty-five years peer harassment in general has also been studied extensively by many developmental and educational psychologists. Specifically, prejudice-based harassment, which includes bullying someone or being bullied because of actual or perceived personal characteristics such as race/ethnicity, has been investigated in multiple studies. The contributions of developmental and educational psychologists should be acknowledged in the paper, and the first sentence of the abstract should be modified. For example, see:

  • Graham, S., & Juvonen, J. (2002). Ethnicity, peer harassment, and adjustment in middle school: An exploratory study. The Journal of Early Adolescence, 22(2), 173-199.
  • Eisenberg, M. E., Neumark-Sztainer, D., & Perry, C. L. (2003). Peer harassment, school connectedness, and academic achievement. Journal of School Health, 73(8), 311-316.
  • Juvonen, J., Nishina, A., & Graham, S. (2000). Peer harassment, psychological adjustment, and school functioning in early adolescence. Journal of Educational Psychology, 92(2), 349.
  • Nishina, A., & Juvonen, J. (2005). Daily reports of witnessing and experiencing peer harassment in middle school. Child Development, 76(2), 435-450.

Moreover, it is worth highlighting the need for a multidisciplinary perspective.

Discussion: Can you suggest any future directions for research in this field?

Best regards and good luck to the authors!

Author Response

Dear Reviewer,

Thank you so much for your thoughtful review of our manuscript! Below, we summarize our responsive edits.

Point 1: Inconsistency in claims between the abstract and main text

Thank you for pointing out the inconsistency between the abstract and main text in terms of which populations were depicted as experiencing higher rates of racial harassment. We have edited the abstract to align with the main text, emphasizing that Black students experience higher rates of racial harassment than students of other backgrounds.

Point 2: Including literature on peer harassment

Thank you for pointing us to four foundational studies that explore the distribution and correlates of peer harassment, and for suggesting we explore studies on prejudice-based harassment. We first incorporated the studies you mentioned in our introduction to acknowledge foundational work on this critical topic. We next searched for, identified, and incorporated a number of studies on race-based harassment and reshaped our abstract, introduction, and conclusion accordingly. Our review of these studies suggests that our research adds to existing research by exploring the distribution of peer racial harassment after major social events (BLM and COVID-19), exploring the mental health correlates of peer racial harassment among new populations, and exploring a novel structural intervention that might reduce student exposure to peer racial harassment in schools.

Point 3: Future directions

Thank you for suggesting that we discuss possible future directions for research. We have added three paragraphs exploring possible future directions. Relatedly, we believe that a major contribution of our work is that it suggests that structural interventions (e.g., educator workforce diversity) might exert schoolwide effects that are particularly beneficial for protecting vulnerable populations from peer racial harassment in schools, and for enhancing their mental health. We therefore suggest other potentially impactful structural interventions that future scholarship could explore. We also note that while our work focuses on school-wide impacts of being in a school with a diverse school mental health workforce, future work could explore how one-on-one therapeutic experiences with race-matched or diverse school mental health providers might confer unique mental health benefits.

We thank you again for your thoughtful feedback, and we hope our revision feels responsive.

With gratitude,

- Authors

Reviewer 2 Report

Comments and Suggestions for Authors

see attached

Author Response

Dear Reviewer,

Thank you so much for your thoughtful review of our manuscript! Below, we summarize our responsive edits.

Point 1: Incorporating a theoretical framework into the introduction, and threading throughout

We appreciate your suggestion that we weave a clear theoretical framework into this article. Our research was guided by a number of theoretical frameworks related to the impacts of school structures, the roles of school mental health providers, and the varied drivers of student mental health. However, we made an intentional choice not to ground our manuscript in theoretical frameworks to avoid flummoxing readers with a lengthy exposition of the varied theories that guided our work. We were guided by five frameworks. The first two were the Systems View of School Climate (Rudasill et al., 2017 -- https://link.springer.com/article/10.1007/s10648-017-9401-y) and a more recently developed comprehensive theoretical framework for mental health promotion in schools (Cavioni, Grazzani, and Ornaghi, 2020 -- https://psycnet.apa.org/record/2020-66286-005). In accord with the first theory, we viewed student behavior (e.g., peer racial harassment) and student mental health (e.g., the development of depressive symptoms) as being a function of structural drivers of school climate, teacher professional development, and student support. And in accord with the second theory, we viewed mental health promotion in schools as being a function of 1) the nature and degree of social and emotional learning opportunities for students and staff (e.g., how school mental health providers help teachers empathize with and provide support to students of varied backgrounds, and how school mental health providers help students empathize with and engage in inclusive practices with students of varied backgrounds); 2) the extent to which school staff are able to help members of marginalized populations develop the kinds of resilience (and sense of identity and support) that allow them to develop in a healthy in way 3) the extent to which school staff are able to create contexts where students are less likely to engage in behaviors that are uniquely harmful to members of marginalized communities, such as peer racial harassment. However, while both popular theoretical frameworks were helpful in our conceptualization of the role of school mental healthcare providers on student behavior and mental health, neither framework provided adequate grounding for our focus on racially minoritized student populations or on the potential impacts of school workforce diversity. These foci reflected key ideas from two other frameworks: intergroup contact theory and critical school psychology theory. Thus, to provide a full and adequate theoretical framework for this research, we felt we would have to explore and synthesize four separate theorical foundations (which could, itself, be the subject of a research project). We felt this would divert substantial attention from the main focus of this piece. Still, it is critical to at minimum pay homage to the brilliant theorists whose work guides ours. We thus now cite these five frameworks in our revision. We hope this discussion provides some context regarding both the theories that undergird our work and the logic behind our decision not to discuss these theories at length in our manuscript, but to instead provide a plain language exposition of the logic model guiding our work.

Point 2: The moderating role of race-matched mental health providers on the link between peer racial harassment and depression

We very much appreciated your suggestion that we explore whether exposure to a race-matched mental health provider negatively moderates the link between exposure to peer racial harassment and developing depressive symptoms. While we believe that such a protective relationship may exist, and feel future research should absolutely explore whether it does, this was not the question guiding our research. We were instead interested in a different moderation question, which we explored within each racial group. For example, with Black students, we asked, “does the relationship between whether a student was racially harassed (0,1) and whether they were in a school with a Black mental health provider (0,1) depend on whether the student is Black?” (and we asked a similar question regarding depressive symptom rates). We use the Jenrich chi squared tests of comparative correlations, but could have just as easily run this as an interacted regression. Indeed, if we do, the results are identical (which makes sense – the same basic conventions apply to both tests). Results for the aforementioned test, conducted as an interacted regression, has the following results:

Racially harassed                 Coefficient           SE           t

Has Black MH Provider        -.006                       .002       -3.05

Black                                      .154                       .003       55.40

Interaction                            -.059                       .006       -9.31

You can view the classic moderation chart here:

https://drive.google.com/file/d/1hFiyUKbgbljGeGJqdGQZO0kOpD-FSVKf/view?usp=sharing

As noted, we were focused on how school mental health providers of various races might broadly influence educator behavior, student behavior, and school climates, in ways that reduce racial harassment and enhance mental health for students of specific races. We theorize this possibility with reference to the aggregate, school-wide impacts that mental health providers have on schools (primarily via their ability to enhance teacher practices by providing professional development, and their ability to change student behavior by providing school-wide social and emotional development instruction). We see our data as being amenable to answering this kind of school-wide effect question.

However, we view the moderation question you pose as being more of a direct student-to-provider level question—in short, that when a student experiences racial harassment, if they are able to interface with a race-matched mental health provider, then the racial harassment will have a smaller negative effect on their mental health. An ideal dataset for exploring this kind of a student-to-provider level moderation would allow researchers to explore how exposure to an incident of racial harassment relates to proximal mental health measures in both of two situations: when they have interfaced with a race-matched mental health provider and when they have not. In short, the data would potentially have within-student variation regarding the kinds of mental health providers they interfaced with following incidents of racial harassment. Our data do not have that level of granularity, and merely allow us to know the racial composition of the mental health providers in the school each student attends (rather than which mental health provider a student interfaced with following any given incident of racial harassment). We also note that if our theory is correct (that being in a school with a race-matched mental health provider reduces exposure to peer racial harassment and reduces depressive symptom rates), then the following model (which we can run with our data, but think we ought not to) might not perform as expected:

DEPRESSIVE SYMPTOMS = a + B1(RACIALLY HARASSED) + B2(RACE MATCHED PROVIDER) + B3(RACIALLY HARASSED x RACE MATCHED PROVIDER)

The reason is that, if our theory is correct (that race matched providers exert schoolwide negative effects on racial harassment and depressive symptoms for minoritized populations), then exposure to “RACE MATCHED PROVIDER” (as measured by our school-level measure) will reduce rates of, and variation in, both RACIALLY HARASSED and DEPRESSIVE SYMPTOMS. Thus, when we compare the relationship between RACIALLY HARASSED and DEPRESSIVE SYMPTOMS for those who did and did not have a PROVIDER IS RACE MATCHED (i.e., those in schools that have a race-matched provider, and those in schools that do not have a race-matched provider), we will be exploring this relationship among students with very different ranges in the other two variables. If we find that the predicted relationship between these variables is different for these two populations, it might simply reflect differences in the ranges of our variables of interest across these populations, rather than differences in the relationships between these variables among students in the two populations. We believe the salve, as noted above, is to leverage student level data for students who have access to, and can interface with, both race matched and non-race matched school mental health providers, and to see if those who did indeed interface with race-matched providers see a stronger relationship between the variables of interest. This would allow us to avoid the possibility that a school-level effect (the constriction of the variance in our exposure and outcome of interest) is washing out the potential to observe variation in the relationship of interest.

In sum, given that our focus here is on the school-level impacts of school mental health providers, and on a different moderation question than the one you pose, and given the limitations of our data, we did not conduct the moderation analysis you suggest here. However, we hope that our discussion of the reasoning behind the analyses we did conduct, and why we did not conduct a moderation analysis, is helpful. And we have included a suggestion that future research should explore this kind of moderation analysis in the Discussion section, as well as discussed the kind of data that we believe would be most amenable to answering this question.

Point 3: Adjustments for multiple tests

We appreciated your suggestion that we indicate whether we made any adjustments for multiple tests (e.g., a Bonferroni correction, which we think would be the most logical adjustment to make given our data and tests). We did not make such an adjustment in this case. In my field (econometrics), we generally make such adjustments in situations where we are evaluating the link between multiple exposures and a single outcome using the same corpus of data, and we make the adjustment to avoid erroneously trumpeting a given exposure as predictive when, indeed, we simply explored relationships between so many exposures and our outcome of interest that a Type I error (detecting an effect that does not exist) became an uncomfortably high probability event. We would make such an adjustment if, for example, we were exploring whether any of ten different exposures (of which peer racial harassment was one) was predictive of depressive symptoms. In each of our tests, we are exploring the relationship between a single exposure and outcome and in some cases, we are exploring whether this relationship is moderated by a single third variable (student race). We do not view this kind of analysis followed by sub-analysis framework as necessitating an adjustment for multiple tests. That said, we note that our p-values for all empirical tests are below 0.001, so, given the small number of tests we are conducting (certainly less than 50), and the very low p values we are seeing in all tests, a Bonferroni correction (alpha_corrected = alpha_prior / number of tests) would not shift our overall determinations about statistical significance. Even assuming we conducted 50 tests (which we do not), our correct alpha would be .05 / 20 = .001), and since all of our p values are below .001, we would still determine that our results are statistically significant in all cases. In terms of how we have rewritten our manuscript, we view the default assumption as being that adjustments for multiple tests were not made. Thus, we believe our methods section as currently written accurately depicts our methods. However, we hope that our discussion clarifies our reasoning behind our methods, and provides reassurance that our results are not merely a reflection of multiple testing error.

Point 4: Clarifying how schools with different numbers of school mental health providers were treated

Thank you for pointing out the lack of clarity regarding how we treated schools with differing numbers of mental health providers. We worried that this might not be clear and have made a full sweep of the article to ensure clarity, including adding additional detail in the methods section regarding how we generated our indicator variables regarding whether each school had a mental health provider of any given race. In essence, we leverage data from the California Survey of School Staff to determine, for each school, whether that school had at least one mental health provider, then leveraged the same data to determine, for each school, whether that school had a mental health provider of a given racial background. If a school had four mental health providers and all were White, then the school would have the following values on the following variables:

Has mental health provider (0,1): 1

White mental health provider (0,1): 1

Black mental health provider (0,1): 0

Asian mental health provider (0,1): 0

Latinx mental health provider (0,1): 0

If a school had two mental health providers, and one was Black and another Asian, then the school would have the following values on these indicator variables.

Has mental health provider (0,1): 1

White mental health provider (0,1): 0

Black mental health provider (0,1): 1

Asian mental health provider (0,1): 1

Latinx mental health provider (0,1): 0

In our analyses predicting the relationship between exposure to peer racial harassment and being in a school that employes a mental provider of a given race, we do not control for the number of mental health providers in the school. We do not control for the number of mental health providers in the school because of the natural collinearity between whether a school employes a mental health provider of a given race and the number of mental health providers they employ.

Point 5: Table one formatting issue

Thank you for pointing out the issue with the formatting of table 1! We are unsure how that formatting error emerged (which did, indeed, make it look like Table 1 was limited to racially harassed individuals). Our guess is that the formatting error emerged as we ported our document into the IJERPH manuscript template. We have corrected this formatting error and also combed the manuscript for others. Fortunately, we did not find any other formatting errors. Thank you again for spotting this!

Point 6: Exploring federal guidance to reduce racial harassment faced by Black students, and discussing current federal context

Thank you for pointing out that our manuscript does not reference federal guidance regarding how to protect Black students from racial harassment. No such guidance currently exists. In addition, thank you for suggesting we point out that the DOE is being restructured, the OCR is hemorrhaging staff, and the OCR is abandoning its long-held mission of protecting students from scholastic racism (in its many forms), and is instead adopting a focus on policing which students play in male and female sports leagues. We have discussed both details (the lack of, and need for, guidance on how to protect Black students from racial harassment, and the need for state education agencies and local education agencies to drive these protective efforts given the reality that these efforts are unlikely to come from the federal government any time soon).

Overall

We want to thank you sincerely for your careful review of our article, and for the many reflections and suggestions you have provided which have strengthened our work. We hope you find our thorough revision to be responsive to your thoughtful review!

- Authors

Round 2

Reviewer 2 Report

Comments and Suggestions for Authors

I appreciate the authors' thoroughness in responding to reviewer feedback. It was helpful. While I understand the authors' rationale about not including the detail about the multiple tests, I would still recommend that they include a sentence or two that provides that rationale. Given that this paper will be read by people of varied disciplines who are likely to have a similar query or concern, including either their rationale or making note that there are no significant changes when adjusting for multiple tests, it would ensure the usability of this paper.

Author Response

Dear reviewer,

Thank you for your input! You may have missed this sentence in the revision (page 8, lines 334-335) which, I believe, is responsive to your request for a sentence indicating that adjustments for multiple tests were not included because they were not deemed necessary:

"Across tests, we do not make a correction for multiple tests because (as seen below) our p values are universally too low for such a correction to augment our findings. "